# Microstructure, Thermal, and Corrosion Behavior of the AlAgCuNiSnTi Equiatomic Multicomponent Alloy

**DOI:** 10.3390/ma12060926

**Published:** 2019-03-20

**Authors:** Eva Fazakas, Bela Varga, Victor Geantă, Tibor Berecz, Péter Jenei, Ionelia Voiculescu, Mihaela Coșniță, Radu Ștefănoiu

**Affiliations:** 1Budapest University of Technology and Economics, Department of Materials Science and Engineering, Bertalan Lajos utca 7, 1111 Budapest, Hungary; efazakas@eik.bme.hu (E.F.); berecz@eik.bme.hu (T.B.); 2Transilvania University of Brașov, Faculty of Materials Science and Engineering, 1 Universității Street, 500068 Brașov, Romania; varga.b@unitbv.ro; 3Politehnica University of Bucharest, Faculty of Materials Science and Engineering, 313 Splaiul Independenței, 060042 Bucharest, Romania; victorgeanta@yahoo.com; 4Loránd Eötvös University, Department of Materials Physics, Pázmány Péter sétány 1/a, 1117 Budapest, Hungary; jenei@metal.elte.hu; 5Politehnica University of Bucharest, Faculty of Engineering and Management of Technological Systems, 313 Splaiul Independenței, 060042 Bucharest, Romania; ioneliav@yahoo.co.uk; 6Transilvania University of Brașov, Faculty of Product Design and Environment, 1 Universității Street, 500068 Brașov, Romania; mihaela.cosnita@unitbv.ro

**Keywords:** high entropy alloy, equiatomic multicomponent alloy, microstructure, corrosion resistance, surface coatings

## Abstract

The paper presents the microstructure and corrosion behavior of an AlTiNiCuAgSn new equiatomic multicomponent alloy. The alloy was obtained using the vacuum arc remelting (VAR) technique in MRF-ABJ900 equipment. The microstructural analysis was performed by optical and scanning electron microscopy (SEM microscope, SEM-EDS) and the phase transformations were highlighted by dilatometric analysis and differential thermal analysis (DTA). The results show that the as-cast alloy microstructure is three-phase, with an average microhardness of 487 HV_0.1/15_. The obtained alloy could be included in the group of compositionally complex alloys (CCA). The corrosion resistance was studied using the potentiodynamic method in saline solution with 3.5% NaCl. Considering the high corrosion resistance, the obtained alloy can be used for surface coating applications.

## 1. Introduction

Over the last decade, the study of multicomponent alloys with several major alloying elements, including high entropy alloys (HEAs) and compositionally complex alloys (CCAs), has been of particular interest for researchers. Since the first articles in the field of high entropy alloys in 2004, their number has increased exponentially to 300 in 2017 [1,2,3]. The interest in these alloys is determined by their special properties, mainly by their high corrosion resistance, as well as by the interest in the development of the general theory regarding the improvement of the properties of alloys both by alloying and by processing those using unconventional technologies.

The alloying process is one of the best methods used for obtaining high values of tensile strength in pure metals. In this case, the enhancement of the properties is based on the deformation, the distortion of the elemental cell as a result of the substitution of the atoms of the base metal with the atoms of the alloying element. The hardening process thus performed is limited by the solubility of the alloying elements in the base metal [4].

This mechanism of increasing the tensile strength also provides the increase of the relative elongation, up to a certain concentration, after which it begins to decrease, due to the occurrence of the intermetallic secondary phases in the structure. If the proportion of the secondary phase increases beyond certain limits, the alloy becomes fragile. Higher proportions of secondary phases are accepted for foundry alloys due to nearing the eutectic concentration in order to provide suitable casting properties.

The substantial enhancement of the strength can be achieved by applying high cooling speeds during solidification, both by finishing the structure, in accordance with the Hall-Petch expression and by creating new non-equilibrium structures (metastable). The extreme of this manner of enhancing the strength is the production of metallic glasses. In this case, the enhancement of the strength characteristics is achieved by freezing the structure corresponding to the liquid state. Thus, by preserving the liquid state structure, characterized by a significantly higher disorder than that of the solid state, the strength characteristics, and not only, are enhanced. The research in this area has reached the stage where amorphous structures can be produced also in the case of blanks with dimensions in the order of millimeters and tens of millimeters, and they are called bulk metallic glasses (BMGs). The shortcoming of obtaining the amorphous structures resides in the size of the processed blanks, which should provide high enough cooling speeds, and in the fact that the amorphizing elements are expensive.

A new direction of development for the improvement of the properties of alloys is the processing of multicomponent, high entropy alloys (HEAs). In the case of HEAs, the aim pursued consists in obtaining structures formed only by solid solutions with elemental cells strongly distorted by the atoms of the alloying elements [5,6].

In this case as well, the aim is to avoid the formation of the secondary, hardening phases, which cause the alloy to become fragile when they exceed a certain proportion. Therefore, in obtaining these materials as well, the cooling rate significantly influences the formation of the desired structure. It is worth mentioning that there are works which demonstrate that the presence of oxide inclusions may have, in fact, beneficial effects in high strength HEAs, additionally strengthened by intermetallic precipitates [7].

Regarding the possibility of directed structure by imposing a targeted chemical composition, it should be noted that in the case of the metallic glasses, the base metal component is present in proportion of 70–90%, in the bulk metallic glasses of 50–70%, and in the HEAs the equimolar composition is recommended, or at least 5% at. These compositions differ significantly from the compositions of the industrial alloys where the base element is predominant.

## 2. Theoretical Analysis 

Since the production of high entropy alloys (HEAs) has a shorter history than the production of metallic glasses and bulk metallic glasses, there are presented some principles related to the formation of the structural constituents in this category of materials [5,6,8].

As known, the state of equilibrium is thermodynamically characterized by the minimum value of the free enthalpy. In the case of an alloy in which in both the solid state and the liquid state we allow the formation of solutions, the free enthalpy of the system (ΔG_mix_) is calculated using the equation:ΔG_mix_ = ΔH_mix_ − TΔS_mix_(1)
where:ΔH_mix_—the enthalpy of the mixture;ΔS_mix_—the entropy of the mixture;T—the temperature.

The entropy of the mixture is the configuration entropy, and it is calculated using the equation:(2)$$\Delta S_{\mathrm{mix}}=-R\sum_{i=1}^{n}X_{i}\ln X_{i}$$
where:R—the universal gas constant, 8.314 J/mol;X_i_—the concentration of the alloying elements, expressed in atomic fractions;n—the number of components.

In the case of an equimolar mixture, Equation (2) is written as: ΔS_mix_ = R × ln(n).

It should be noted that in the case of a ternary, equimolar alloy, the mixing entropy exceeds the melting entropy of most metals. In other words, the disorder created by alloying three metals is comparable to the disorder caused by melting a metal.

As an example, the entropy variance when melting aluminum is:(3)$$\Delta S_{\mathrm{top}}=\frac{\Delta H_{\mathrm{top}}}{T_{\mathrm{top}}}=\frac{10457}{933}=11.21\;J/\mathrm{mol}\cdot K$$

When forming a melt with three alloying elements in equimolar concentrations, the configuration entropy is: (4)$$\Delta S_{\mathrm{mix}}=-R\sum_{i=1}^{3}X_{i}\ln X_{i}=-8.314\ln(0.33)=9.15\;J/\mathrm{mol}\cdot K$$

The alloys with 5 or 6 alloying elements in equimolar concentrations already have very high entropies. These alloys are called “high entropy alloys” (HEAs).

In the case of multicomponent alloys, the mixing enthalpy is calculated using the expression:(5)$$∆H_{\mathrm{mix}}=\sum_{i=1,i\neq j}^{n}\Omega_{ij}X_{i}X_{j}$$
where: Ω_ij_ is the parameter of interaction between the components i and j in the case of regular solutions, calculated using the expression known as Miedema’s rule:(6)$$\Omega_{\mathrm{ij}}=4\times \Delta H_{AB}^{\mathrm{mix}}$$

For binary systems, the mixing enthalpy values ($H_{AB}^{\mathrm{mix}}$) are presented in the literature [6,8,9,10].

Furthermore, the Hume-Rothery rules provide semi-quantitative information regarding the probability of formation of the different phases (solid solutions/intermediate phases/amorphous structure).

(1) The relative difference between the diameters of the alloying elements atoms (δ) is calculated using the expression:(7)$$\delta =100\cdot \sqrt{\sum_{i=1}^{n}X_{i}(1-\frac{r_{i}}{\bar{r}}}{)}^{2}$$
where: (8)$$\bar{r}=\sum_{i=1}^{n}X_{i}\cdot r_{i},$$
and r_i_—the atomic radius for the component ‘i’.

The processing of a large number of experimental results led to the establishment of the following rules regarding the nature of the structures obtained [6,7,8,9,10]: solid solution structures are formed when, −22 ≤ ∆H_mix_ ≤ 7 kJ/mol and 0 ≤ δ ≤ 8.5; in the case of bulk metallic glasses −35 ≤ ∆H_mix_ ≤ −8.5 kJ/mol and δ ≥ 9. 

(2) The electronegativity difference (Δχ), defined by the expression:(9)$$\Delta \chi =\sqrt{\sum_{i=1}^{n}X_{i}{\left(\chi_{i}-\bar{\chi }\right)}^{2}}$$
where χ_i_ is the Pauling electronegativity for the element “i”.

(3) The valence electron concentration (VEC), calculated using the expression:(10)$$\mathrm{VEC}=\sum_{i=1}^{n}X_{i}{\left(\mathrm{VEC}\right)}_{i}$$
where (VEC)_i_ is the VEC for the alloying element “i”.

If VEC ≤ 6.87, the alloy will have the structure bcc, if VEC ≥ 8, the fcc structure is formed, and if the VEC parameter is between these two values, the alloy will have the (bcc + fcc) structure.

In order to assess the ability to form the solid solution, the Ω parameter was also introduced, defined by the expression:(11)$$\Omega =\frac{T_{m}\cdot \Delta S_{\mathrm{mix}}^{}}{\left|\Delta H_{\mathrm{mix}}\right|}$$

The melting temperature for the alloy, in Kelvin degrees, is calculated using the additivity rule: (12)$$T_{m}=\sum_{i=1}^{n}X_{i}{\left(T_{m}\right)}_{i}$$

Until now, in the case of high entropy alloys, the study of magnetic, elasticity, and corrosion-resistance properties prevails [1,2,3,11,12,13,14,15,16,17]. The paper [18] analyses the properties of the AlCuAgTi system high entropy alloy proposed to be used for metal brazing.

The obtainment and analysis of the microstructure and properties of the new AlAgCuNiSnTi multicomponent alloy in equal atomic concentrations (equimolar) are proposed in this paper. Fifteen binary systems can be formed between the six chemical elements used for this alloy. Of these systems, only in a (Cu-Ni) system do the two elements exhibit unlimited mutual solubility in solid state; in a (Ni-Ag) system, the elements exhibit limited solubility in liquid state and they are completely immiscible in solid state; in an (Al-Sn) system, the elements exhibit unlimited solubility in liquid state and they are completely insoluble in solid state; in a (Cu-Ag) system it is formed eutectically between the two marginal solutions, and in the remaining 11 binary systems, besides the solid marginal solutions and/or the solidified elements in pure state (due to the presence of an eutectic transformation), a series of intermetallic compounds with fixed composition and stable compounds form on a narrow range of composition

## 3. Materials and Methods 

A multicomponent alloy, consisting of six chemical elements in equal atomic concentrations, was made of pure elements in a Vacuum Arc Remelting (VAR) equipment type MRF-ABJ900 (Materials Research Furnaces INC., Suncook, NH, USA), with a non-consumable tungsten–thorium electrode under an argon-protective atmosphere. For this purpose, after obtaining a vacuum of 5 × 10^−4^ mbar, the work chamber was flooded with 5.3 pure Argon for 20 min to ensure the stability of the electric arc formed between the tungsten–thorium electrode and the metallic load.

The homogeneity of the sample was ensured by remelting it (five times). During the melting and remelting operations, the metal losses due to burning–volatilization were insignificant. The checking was performed by comparing the weight of the solid metal load (100 g) and of the ingot obtained after melting-remelting (99.67 g).

Some physical data and proportion (in atomic and weight percent) of each chemical element of the new alloy are shown in Table 1.

Using the data presented in literature [5,6,9,10], the thermodynamic parameters have been calculated based on the Equations (2) and (5)–(12), the obtained values being given in Table 2.

For the microstructural analysis, the samples were grinded using sandpapers with different grit sizes (80–1200) and then polished with alumina (Al_2_O_3_) in suspension. The phases that are presented in the microstructure show adequate contrast without metallographic chemical attack. The microstructures of the samples were examined by scanning electron microscopy (using a FEI Inspect S microscope (Hillsboro, OR, USA) equipped with EDAX system Z2e for chemical composition estimation, EDAX Inc., Mahwah, NJ, USA) and X-ray diffraction (using a Rigaku - SmartLab diffractometer, having CuKα radiation, Rigaku Europe SE, Neu-Isenburg, Germany). 

## 4. Results

The results of the calculations made using Equations (5) and (7) for the composition proposed for analysis confirmed that compositionally the new alloy only partially meets the high entropy alloy criterion, because: ∆H_mix_ = −10.13 kJ/mol, and δ = 8.92. With these values for the parameters ΔH_mix_ and δ, the alloy can be placed in the range of intermediate phase alloys [8]. The electronegativity of the alloy cannot be considered as a criterion for the type of structure being formed. The thermodynamic values obtained falls within the range of the values characteristic for alloys with solid solution structures [6], having intermetallic phases and amorphous phases. Interestingly, the value of the VEC parameter for the analyzed alloy is at the limit between the bcc and fcc solid solution ranges, i.e., VEC ≥ 8 favors the formation of solid solutions with fcc, while VEC < 6.87 favors the formation of solid solutions with bcc [6]. According to the values of the parameters Ω and δ, the analyzed alloy is placed in the range of alloys with intermediate structure [5].

The actual density of the alloy determined by hydrostatic weighing (Table 2), is lower than the theoretical value calculated additively (VEC), column 7. This difference can be explained by the lower compactness of the alloy, the formation of some light intermetallic phases and very small size micro-shrinkages (Figure 1).

The microstructure of the cast ingot (Figure 2a) is quite homogeneous and consists of three phases. It should be noted that the structure of the cast alloy contains, besides the three main phases, some compounds like Chinese letters, visible in the bright phase. 

During quick solidification (melt spinning method) can be obtained ≈1 mm thick strips, whose microstructure seems to be more homogenous and grain refined (Figure 2b).

The stability of the as-cast microstructure was tested by homogenization heat treatments at a temperature of 800 °C. The representative microstructure resulting is shown in Figure 2c. It should be noted that both the microstructure of the strip and of the heat-treated (homogenized) sample contains only the three main phases that were identified using X-ray diffraction.

The X-ray diffractogram of the as-cast equimolar AlAgCuNiSnTi high entropy alloy is shown in Figure 3.

The phases occurring in the multi-component alloy have been identified in all three processing stages (as cast sample, thin cast strip, and heat treated) both by diffraction and differential thermal analysis (DTA) It has been noted that the Ag_3_Sn phase represents the dendritic matrix in which the other phases are present, having different dimensions and morphologies. Thus, the TiNi_1.056_Sn phase is in the form of dark compounds, surrounded by the Sn_5_Ti_6_ phase, having the appearance of gray islands with different sizes (11 to 30 μm) and different shapes (rounded or elongated). The Al_1.33_Cu_0.67_ phase has well-defined shapes, with quasi-rectangular faces, and average dimensions between 4 and 10 μm.

The results of the X-ray diffraction analysis highlighted the existence of four phases in the structure of the multicomponent alloy.

In the following the Powder Diffraction File (PDF) Card number and the measured lattice constants are also given in addition to the chemical composition. The main phase with the TiNiSn (PDF Card No.: 01-080-2880) composition crystallizes in an fcc system (a = 0.5936 ± 0.0008 nm) and is a semi-Heusler compound [19], forming elongated separations with rounded edges; the interdendritic space phase with the Ag_3_Sn (PDF Card No.: 01-084-8210) composition crystallizes in the orthorhombic system (a = 0.4773 ± 0.0005 nm, b = 0.6008 ± 0.0007 nm, c = 0.5155 ± 0.0006 nm), has a shiny appearance, and the third, darker colored phase, with the Al_1.33_Cu_0.67_ (PDF Card No.: 00-067-0160) composition crystallizes in a simple cubic system (a= 0.2906 ± 0.0006 nm), occurring in the form of grey ink stains (Figure 2a,c). Beside this a small amount of hexagonal Sn_5_Ti_6_ phase (PDF Card No.: 01-072-3255) was identified in the alloy (a = 0.9202 ± 0.0009 nm, c = 0.5673 ± 0.0007 nm).

Both the homogeneity of the structure and the existence of the phases in the structure are confirmed by the spectrum of elemental distribution of the alloying elements in the three samples (Figure 4).

It is to be noticed that in the case of the strip sample, Ti-Ni-rich phases were formed (Figure 4b).

The mechanical characteristics were assessed by determining the microhardness using an FM-700 AHOTEC tester (Future-Tech Corp, Talkpier Kawasaki, Kanagawa, Japan). The results obtained for the three samples are shown in Figure 5. For the cast and the homogenized samples, the microhardness was determined separately for each phase. Due to the finer structure of the cast strip, only the global microhardness of this sample was determined.

The microhardness values obtained for the analyzed multicomponent alloy are lower than those considered to be characteristic of high entropy alloys [20,21,22,23].

The thermal analysis was performed using the dilatometer L75/230-LINSEIS (DIL-Linseis Messgeräte Gmbh, Selb, Germany) and STA-449F3 Jupiter (Netzsch-Geratebau GmbH, Selb, Germany). The DIL (Dilatometer) and DTA (Differential Thermal Analysis) results are presented in Figure 6. The curves of the DTA analysis during heating and cooling for the cast alloy are shown in Figure 6a. The expansion curves for the alloy obtained in the MRF-ABJ900 equipment (Materials Research Furnaces INC., Suncook, NH, USA) and the homogenized one, indicating the initial lengths (L0) for the specimens used are shown in the Figure 6b. In both analysis methods, the heating and cooling rates were the same, 10 °C/min.

The DTA and DIL curves show the partial melting of the easy fusible phases (Al_1.33_Cu_0.67_ and Ag_3_Sn). The diffusion phenomena between solid and liquid phases contribute to fragmentation of the Al_1.33_Cu_0.67_ phase and homogenization of the Ag_3_Sn phase by the disappearance of the phases separated in the form of Chinese letters. Due to these phenomena the delineation of the phases is more evident in the homogenized sample (Figure 4c), if in the molded sample (Figure 4a). The stability of the structure results both by the DTA analysis and dilatometric measurements, consistent with the microstructures shown in Figure 2a,c.

The corrosion behavior was studied in a 3.5% NaCl solution, using a BioLogic SP-150 potentiometer and a 3-electrode cell (Bio-Logic Science Instruments, Seyssinet-Pariset, France). The Tafel diagrams recorded for different corrosion intervals are shown in Figure 7.

The corrosion resistance was assessed by calculating the corrosion rate (v_cor_) using the expression:(13)$$v_{\mathrm{cor}}=K_{1}\frac{i_{\mathrm{cor}}}{\rho }\mathrm{EW}$$
where:ρ is the density in g/cm^3^the constant *K*_1_ = 3.27 × 10^−3^, (mm·g/μA·cm·year),i_cor_—the corrosion current, (μA/cm^2^), obtained by graphical construction from the Tafel semilogarithmic polarization curve.EW—equivalent weight, calculated using the expression:
(14)$$EW=\frac{1}{\sum \frac{n_{i}\cdot f_{i}}{M_{i}}}$$

For the alloy studied, EW = 6.88.

The results of the calculations are shown in the Figure 8.

The evolution of the corrosion resistance indicate an initially high corrosion rate, that decreases very quickly due to passivation of the alloy surface, and stabilizes at a value of 0.004 (mm/y (millimeters per year)).

The obtained corrosion rate for the studied alloy is better than other HEAs. For FeNiCoMnCrAl and FeNiCoMnCr high entropy alloys, the reported corrosion rate in a 3.5% NaCl solution was 0.009 mm/y and 0.226 mm/y (millimeters per year), respectively [24]. For the commercially available Hastelloy C276 (UNS N 10276), the obtained corrosion rate, for the same conditions, is 0.009 mm/y. The obtained corrosion rate was higher than the CR obtained for FeNiCoMnCrNb microalloyed with Y (0.002 mm/y). On the other hand, the corrosion rate of the new obtained material can be also compared with stainless steel. The corrosion rate of stainless steel (AISI 316L) in sodium chloride solution with concentration of (5 g/L = 0.5%) was 0.00185 mm/y [24]. The corrosion rate of our alloy in the 3.5% (35 g/L) sodium chloride was 0.004 mm/y. Even if the corrosion rate of the new alloy is worse than the corrosion rate of the steel (because of the concentration of the sodium chloride solution), the obtained results indicates that the alloy exhibits a good corrosion resistance in the solution of 3.5% NaCl [24].

## 5. Discussion

Although the alloy examined contains six different alloying elements in equimolar concentrations, the calculations and microstructural aspects indicate there is no high entropy. By calculating the empirical parameters used to assess the microstructure of a multicomponent alloy, the conclusion drawn is that it does not meet the criteria for high entropy alloys. On the basis of these criteria, this alloy is placed in the range of intermediate phase alloys.

The structural studies performed by microscopy and roentgenogram confirm that the new material analyzed does not accomplish all the rules to be considered as a high entropy alloy because it contains several phases, i.e. four. The phases do not have a dendritic appearance, they are not solid solutions, and they form intermediary phases with cubic, tetragonal, and orthorhombic structures.

The homogeneous structure of the multicomponent alloy is confirmed by the SEM-EDX analysis results. The main phase TiNiSn crystallizes in the cubic system based on the semi-Heusler compound.

The phases exhibit very different hardness values, the maximum value of 600 HV_0.1/15_ being measured for TiNi_1.056_Sn phase.

The microstructure obtained is stable; the thermal analyses do not highlight significant structural changes during heating at 800 °C. The increase in the cooling speed, by the casting of strips, has a significant effect on both the grain refinement and the homogeneity of the structure.

The corrosion measurements have highlighted the positive action of the titanium oxide on the corrosion rate value.

Considering its high corrosion resistance, the obtained alloy can be used for surface coatings applications.

## 6. Conclusions

The findings of the study can be summarized as follows:(1)The theoretical calculations for the determination of the enthalpy and entropy values for the analyzed alloy (ΔH_mix_, ΔS_mix_, δ) were validated by the microstructural analyzes (X-ray diffraction and SEM microscopy), with the result that the studied alloy does not meet all the conditions for be considered a high entropy alloy, but only a multi-component alloy;(2)Each phase in the alloy exhibits different values of microhardness, and by applying the homogenization treatment there was a slight decrease in hardness for all phases;(3)The heat treatment for homogenization realized at 800 °C did not cause major changes in the alloy microstructure;(4)The alloy has the ability to withstand the corrosive attack by forming a passivation layer (about 60 min), the corrosion rate being subsequently very low (0.004 mm/y).

## Figures and Tables

**Figure 1 materials-12-00926-f001:**
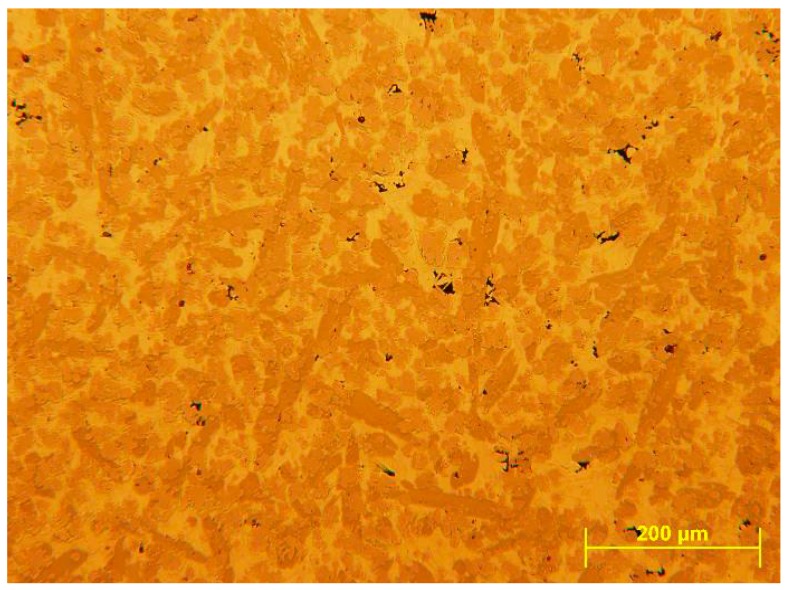
Microstructure of the cast ingot with micro-shrinkages.

**Figure 2 materials-12-00926-f002:**
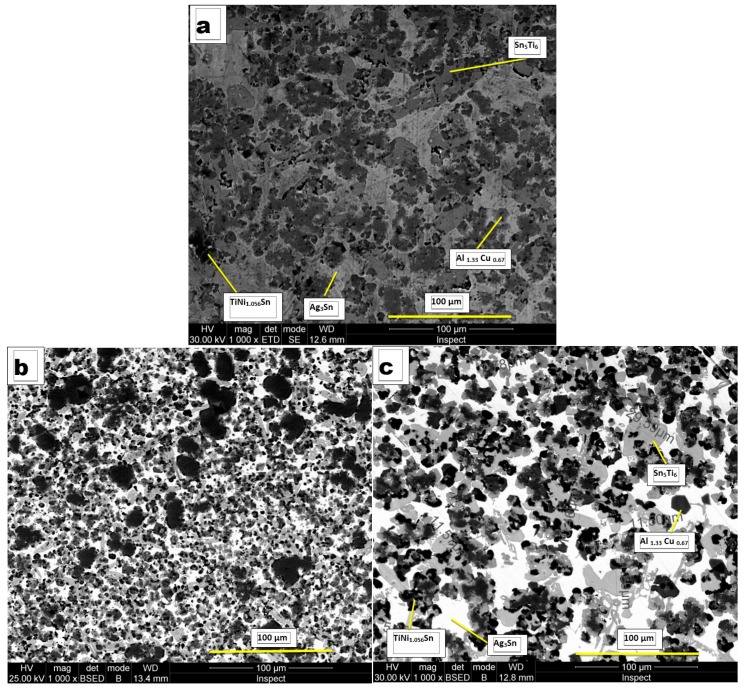
Microstructure of the equimolar AlAgCuNiSnTi alloy: (**a**) cast ingot, 50 mm diameter, 10 mm thick; (**b**) ≈1 mm thick cast strip; (**c**) sample homogenized at 800 °C.

**Figure 3 materials-12-00926-f003:**
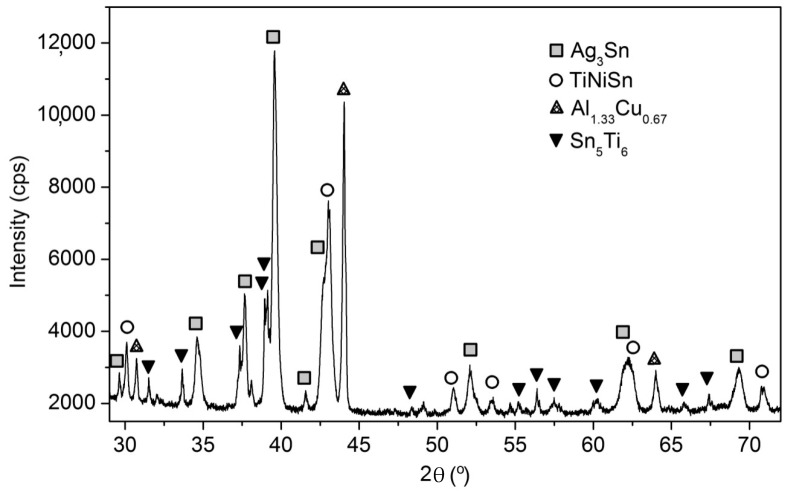
A part of the X-ray diffractogram of the equimolar AlAgCuNiSnTi alloy, as-cast.

**Figure 4 materials-12-00926-f004:**
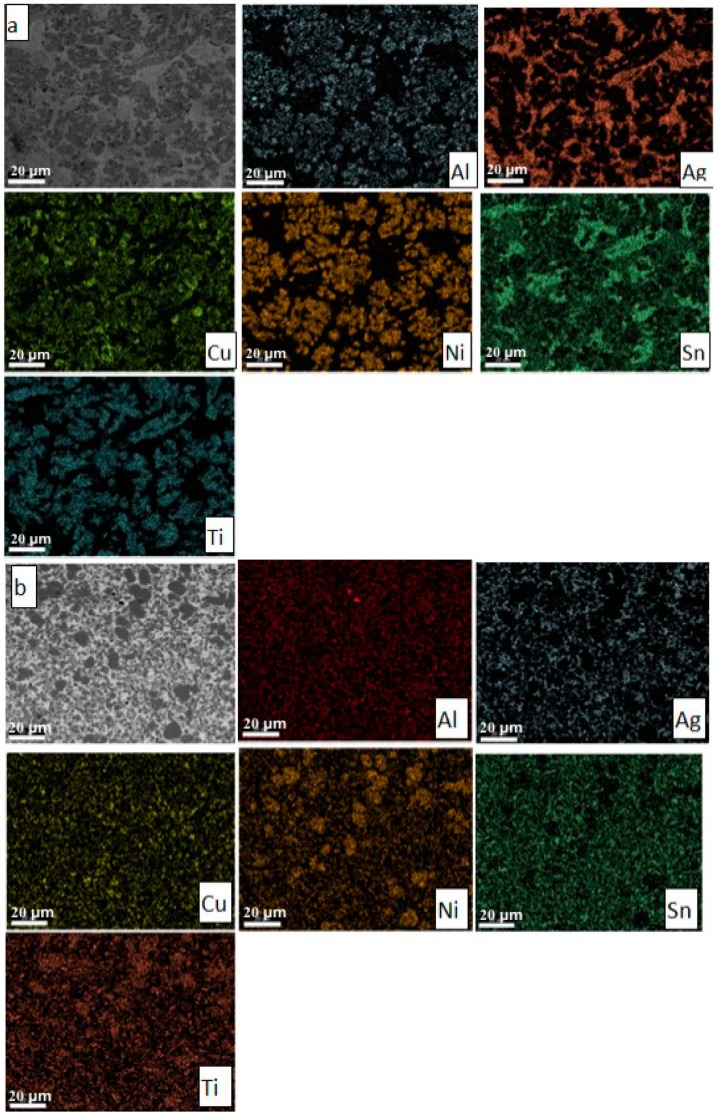
SEM analysis for the equimolar AlAgCuNiSnTi alloy. Electron microscopy image of selected area of the alloys and the spectrum distribution of the alloying elements in the sample: (**a**) as cast specimen obtained by melting in electric arc; (**b**) strip sample; (**c**) alloy homogenized at 800 °C.

**Figure 5 materials-12-00926-f005:**
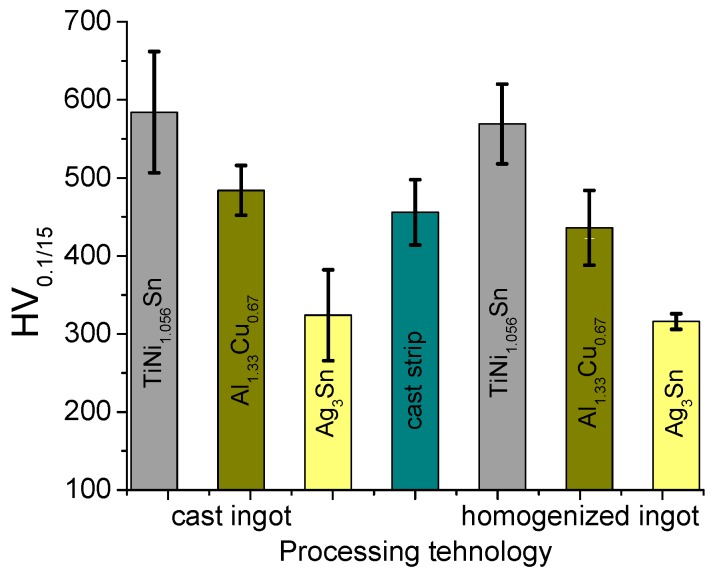
Microhardness values of different samples of the equimolar AlAgCuNiSnTi alloy.

**Figure 6 materials-12-00926-f006:**
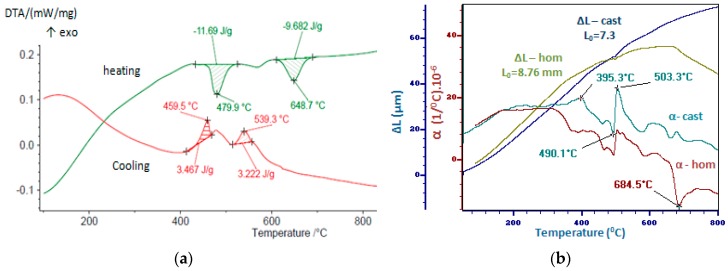
Thermal analysis results: (**a**) DTA curves during heating and cooling; (**b**) expansion curves (ΔL) and variation of the physical expansion coefficient according to the temperature (α).

**Figure 7 materials-12-00926-f007:**
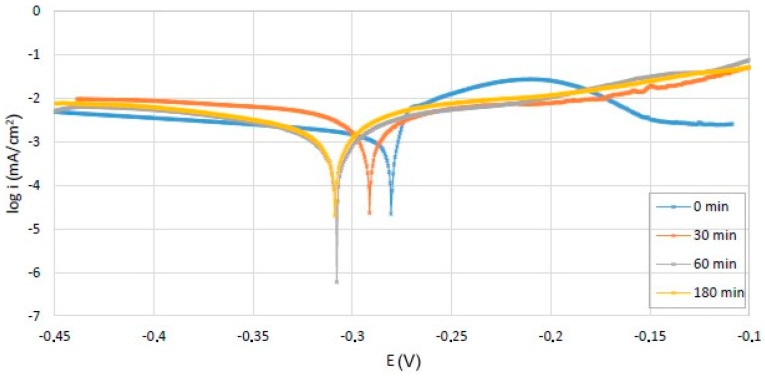
Tafel diagrams for corrosion in solution of 3.5% NaCl for 0, 30, 60, and 180 min.

**Figure 8 materials-12-00926-f008:**
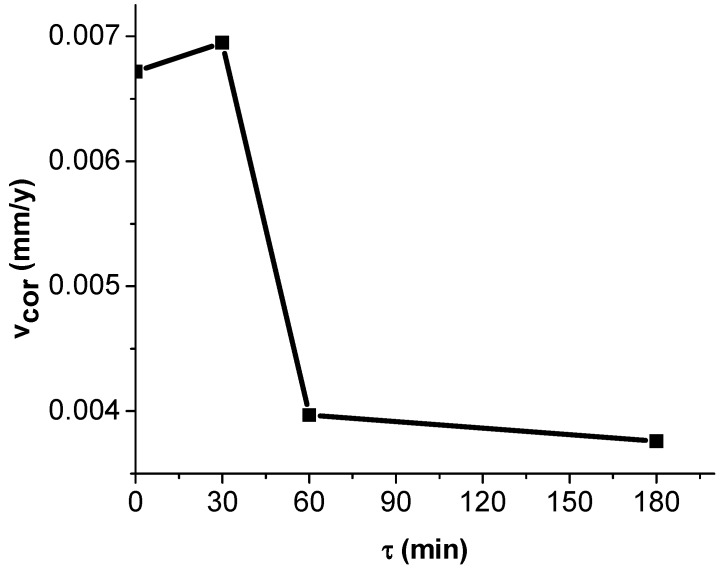
The evolution of corrosion resistance over time for the multicomponent equimolar AlAgCuNiSnTi alloy.

**Table 1 materials-12-00926-t001:** Some physical data and the proportion of chemical elements that form the AlAgCuNiSnTi alloy.

Element	Atomic Ratio (%)	Weight (%)	M (g/mol)	r (Å)	n	Pauling Electronegativity	VEC	ρ (g/cm^3^)	t_m_ (°C)
Al	16.6	6.38	26.98	1.432	3	1.61	3	2.7	660
Ag	16.6	25.52	107.87	1.445	1	1.93	11	10.5	960
Cu	16.6	15.03	63.55	1.278	2	1.90	11	8.96	1083
Ni	16.6	13.89	58.69	1.246	2	1.91	10	8.9	1440
Sn	16.6	27.91	118.71	1.620	4	1.96	4	7.26	232
Ti	16.6	11.25	47.87	1.462	4	1.54	4	4.5	1668
∑	100	100						-	

Remark: M—molecular mass; n—valence; ρ—density of components; t_m_—melting temperature.

**Table 2 materials-12-00926-t002:** Thermodynamic data calculated for the AlAgCuNiSnTi equimolar alloy.

Property	ρ (g/cm^3^)		∆H_mix_ (kJ/mol)	δ	ΔS_mix_ (J/mol·K)	Δχ	VEC	Ω
	Theoretical	Actual						
Value	7.967	6.654	−10.13	8.92	14.9	0.178	6.88	1.81

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
