# Peer review of "Microstructure, Thermal, and Corrosion Behavior of the AlAgCuNiSnTi Equiatomic Multicomponent Alloy"

_materials, 2019, doi:10.3390/ma12060926_

Reviewer 1 Report

This paper reports the microstructure, thermal and corrosion behavior of new multicomponent alloy AlAgCuNiSnTi. The microstructure study has detected the three phases in the new alloy. The average microhardness is 487 HV. It is interesting finding that the alloy has the high corrosion resistance and can be used for the surface coatings applications.

The paper is well organized and has revealed that AlAgCuNiSnTi is the new multicomponent alloy showing the high corrosion resistance. So I believe the manuscript meets all criteria necessary for Materials. But, before the acceptance, I recommend the authors to address the comments listed below.

(1)    I recommend the authors to add a comparison of microhardness between AlAgCuNiSnTi and several HEA.

(2)    I recommend the authors to add a brief survey of papers, reporting multicomponent alloys showing the high corrosion resistance. I think it is useful for the general readers.

Author Response

Dear reviewer,

Thank you very much for reading the manuscript and for giving valuable comments and suggestions. We have revised it accordingly and detailed corrections are given in the attached revised version of our manuscript (with red are marked the changes/additions).

Reviewer 2 Report

1. Does any reference support the equation (12)? The melting point of NiAl is 1638°C, but the melting points of Ni and Al are 1455°C and 660°C, respectively. Therefore, this equation is not applied in NiAl alloy. The melting points of alloys depend on their bonding types and bonding energies, not a simple formula. The followed calculations based on this wrong assumption are thus not reliable.

2. The phases of this alloy are identified not accurately. Many peaks of the XRD pattern (Figure 3) are not marked.

3. What are the compositions of the three phases in this alloy, C, OR and T. Also, what are their lattice constants? They should be mentioned in the manuscript.

4. What kind of cubic system of the Phase C (line 212), BCC, FCC, DC or SC? This mention is too rough.

5. This manuscript has no conclusions.

From above questions, I do not think it is a finished study.

Author Response

(The authors gave the same response as above.)

Reviewer 3 Report

The authors prepared a new HEA by means of VAM and casting. The article presents some interesting results. However, the quality has to be extensively improved. Coments:

1) improve the quality of English language - it is very very poor at some places e.g. "The alloying process was between the first methods used for obtain the enhancement" words between and for are completely wrong. Or here "In what regards the possibility of directing the structure by choosing the chemical composition, 75 it should be noted that in the case "- this does not even make sense. Maybe check by professional proofreading company would be in place.

2) "The substantial enhancement of the mechanical properties, and not only, can be achieved by  applying high cooling speeds during solidification, both by finishing the structure, in accordance  with the Hall-Petch expression" - This statement is written many times in the paper and it is wrong. The improvement of strength and hardness does not mean mechanical properties are improved since term mechanical properties encompasses also ductility, fracture resistance etc... 

Instead of "mechanical properties", write everywhere "strength"

3) This claim: "In this case as well the aim is to avoid the formation of the secondary, hardening phases, which cause the alloy to become fragile when they exceed a certain proportion. Therefore, in obtaining these materials as well, the cooling rate significantly influences the formation of the desired  structure" is not true since the HEA concept has been extended and many new multi-phase HEAs with exceptional properties were prepared  for instance see this reference -https://doi.org/10.1016/j.scriptamat.2018.07.034. You can include it in introduction. 

4) Instead of using T, C and OR for designation of phases, just put their real names to all figures. It improves the clarity for readers.

5) Improve the validity of the terms. For example images in Figure 4 are done by "Electronic microscopy". The name of the method is "Electron microscopy" right? Spell check of MS Word can be wrong sometimes

6) The  presented scale bars are too small and not readable in most SEM micrographs.

7) provide point EDS analysis results for respective phases in the microstructures (average of several points of same phae). It is not clear how exactly were the the chemical composition for XRD like "TiNi10.56Sn or TiNiSn" determined. The chemical composition given by ICSD database from XRD softwares does not need to reflect the real phases.

8)Explain, how was the composition of the alloy designed?

9) The corrosion data has to be compared to some established materials or other HEAs and extensive discussion has to be prided.

Author Response

Dear reviewer,

Thank you very much for reading the manuscript and for giving valuable comments and suggestions. We have revised it accordingly and detailed corrections are given in the attached revised version of our manuscript (with red are marked the changes/additions).

Round  2

Reviewer 2 Report

I agree with this version of manuscript.

Author Response

Dear reviewer,

Thank you very much for reading the manuscript and for giving valuable comments and suggestions. We have revised it accordingly and detailed corrections are given in the attached revised version of our manuscript (with red are marked the changes/additions).

We have made an extensive review of English language (with a help of a colleague working in U.S.). Some of the paragraphs were modified according to the review suggestions and are provided in the new version of the manuscript.

Reviewer 3 Report

There is still several points to be improved:

1) The presented scale bars are STILL TOO SMALL  and not readable. It is impossible to see the numbers of the scale bars. 

2) Again my question is - you performed EDS mappong - so why dont you present any point analysis of the respective phases? Also connect them to XRD results. 

3) Provide extensive comparisson of your corrosion results with other data - present concrete results and numbers This is too general and really does not show anything with respect to your alloy:"It has been found that aluminum causes the reduction in corrosion resistance in 275 acidic media. The tests performed showed that the pitting potential of alloys AlxCrFe1.5MnNi0.5 276 was significantly lower than that of CrFe1.5MnNi0.5 alloy without Al [25]. The single-phase HEA 277 structures, especially the FCCs, have superior corrosion resistance compared to multiphase 278 structures. For example, the CoCrFeNi alloy exhibits much better corrosion resistance than 279 conventional alloys [26]

Also include reply 9 in the paper. "Answer 9. Thank you very much. We agree with your remark. We think the best comparison material is the stainless steel. The corrosion rate of stainless steel (AISI 316L) in sodium chloride solution with concentration of (5g/liter= 0,5%) was 0.00185 mmpy. The corrosion rate of our alloy in the 3,5% (35g/liter) sodium chloride the corrosion rate was 0.004 mmpy."

It is OK to present results, even when its worse that steel.  

Author Response

Dear reviewer,

Thank you very much for reading the manuscript and for giving valuable comments and suggestions. We have revised it accordingly and detailed corrections are given in the attached revised version of our manuscript (with red are marked the changes/additions).

Reply to the Reviewer's comments (the comments are in normal style while the answers are in bold)

1) The presented scale bars are STILL TOO SMALL and not readable. It is impossible to see the numbers of the scale bars.

Answer 1. We have increased the resolution and zoom of the SEM micrographs and the scale bars are easily readable now.

2) Again my question is - you performed EDS mappong - so why dont you present any point analysis of the respective phases? Also connect them to XRD results.

Answer 2. Thank you for your comment. We have made changes accordingly. We have made extensive comments regarding the analysis of the presented phases, including EDS mapping and XRD results.

3) Provide extensive comparisson of your corrosion results with other data - present concrete results and numbers This is too general and really does not show anything with respect to your alloy:"It has been found that aluminum causes the reduction in corrosion resistance in 275 acidic media. The tests performed showed that the pitting potential of alloys AlxCrFe1.5MnNi0.5 276 was significantly lower than that of CrFe1.5MnNi0.5 alloy without Al [25]. The single-phase HEA 277 structures, especially the FCCs, have superior corrosion resistance compared to multiphase 278 structures. For example, the CoCrFeNi alloy exhibits much better corrosion resistance than 279 conventional alloys [26]. Also include reply 9 in the paper. "Answer 9. Thank you very much. We agree with your remark. We think the best comparison material is the stainless steel. The corrosion rate of stainless steel (AISI 316L) in sodium chloride solution with concentration of (5g/liter= 0,5%) was 0.00185 mmpy. The corrosion rate of our alloy in the 3,5% (35g/liter) sodium chloride the corrosion rate was 0.004 mmpy." It is OK to present results, even when its worse that steel.  

Answer 3. Thank you for your valuable idea. We have made a comparison of the corrosion results of our alloy with other materials (HEA’s, Hastelloy and steel). We have also included in the paper the reply 9.
